# Potential of training of anti-*Staphylococcus aureus* therapeutic phages against *Staphylococcus epidermidis* multidrug-resistant isolates is restricted by inter- and intra-sequence type specificity

Camille Kolenda,[1,2] Mélanie Bonhomme,[1,2] Mathieu Medina,[1,2] Mateo Pouilly,[1] Clara Rousseau,[1] Emma Troesch,[1] Patricia Martins-Simoes,[1,2] Marc Stegger,[3,4] Paul O. Verhoeven,[5,6,7] Floriane Laumay,[1,2,8] Frédéric Laurent[1,2,8]

**ABSTRACT** Phage therapy appears to be a promising approach to tackle multidrug-resistant bacteria, including staphylococci. However, most anti-staphylococcal phages have been characterized in *Staphylococcus aureus*, while a limited number of studies investigated phage activity against *S. epidermidis*. We studied the potential of phage training to extend the host range of two types of anti-*S. aureus* phages against *S. epidermidis* isolates. The Appelmans protocol was applied to a mixture of *Kayvirus* and a mixture of *Silviavirus* phages repeatedly exposed to seven *S. epidermidis* strains representative of nosocomial-associated sequence types (ST), including the world-wide disseminated ST2. We observed increased activity only for the *Kayvirus* mixture against two of these strains (ST2 or ST35). Phage subpopulations isolated from the training mixture using these two strains (five/strain) exhibited different evolved phenotypes, active only against their isolation strain or strains of the same ST. Of note, 16/47 ST2 strains were susceptible to one of the groups of trained phages. A comparative genomic analysis of ancestral and trained phage genomes, conducted to identify potential bacterial determinants of such specific activity, found numerous recombination events between two of the three ancestors. However, a small number of trained phage genes had nucleotide sequence modifications impacting the corresponding protein compared to ancestral phages, two to four of them per phage genome being specific of each group of phage subpopulations exhibiting different host range. The results suggest that anti-*S. aureus* phages can be adapted to *S. epidermidis* isolates but with inter- and intra-ST specificity.

**IMPORTANCE** *S. epidermidis* is increasingly recognized as a threat for public health. Its clinical importance is notably related to multidrug resistance. Phage therapy is one of the most promising alternative therapeutic strategies to antibiotics. Nonetheless, only very few phages active against this bacterial species have been described. In the present study, we showed that phage training can be used to extend the host range of polyvalent *Kayvirus* phages within the *Staphylococcus* genera to include *S. epidermidis* species. In the context of rapid development of phage therapy, *in vitro* forced adaptation of previously characterized phages could be an appealing alternative to fastidious repeated isolation of new phages to improve the therapeutic potential of a phage collection.

**KEYWORDS** *Staphylococcus aureus*, *Staphylococcus epidermidis*, bacteriophages, phage training, host range extension, adaptation, evolution, DNA recombination, *Silviavirus*, *Kayvirus*

B acteria of the genus *Staphylococcus* are both dominant commensal microorganisms of the skin microbiome and major human pathogens. Although *Staphylococcus*

Address correspondence to Camille Kolenda, camille.kolenda@chu-lyon.fr.

The authors declare no conflict of interest.

See the funding table on p. 13.

*aureus* is the main life-threatening species (1), *S. epidermidis* is increasingly recognized as a threat, especially in the context of nosocomial and device-associated infections. It is notably responsible for up to 30% of nosocomial bacteremia and prosthetic joint infections (2, 3). The clinical importance of *Staphylococcus epidermidis* is related primarily to its frequent multi-resistance to antibiotics, especially methicillin resistance that is typically detected in more than 70% of the strains and can reach up to 90% of strains (4). Furthermore, its ability to form biofilms over various surfaces, in which bacteria are protected from antibiotics and the immune system, is a key trait of the species as well as a determinant of its pathogenicity (5). It is also of note that some sequence types (STs) are over-represented in infections, such as the most prominent lineage ST2 that has disseminated worldwide in the hospital environment and exhibits increased virulence and resistance compared to commensal strains (6, 7).

In this context, there is an urgent need for alternative strategies to antibiotics. Phage therapy, which relies on the use of bacteria-specific viruses called bacteriophages (or phages), is one of the most promising approaches in several aspects. First, as phages target distinct receptors from antibiotics and have different mechanisms of antibacterial activity, they can kill antibiotic-resistant bacteria but can also be used to obtain synergistic effects with antibiotics (8). Second, due to the production of specific enzymes such as depolymerases and lysins allowing penetration through the biofilm matrix, phages have the potential to eradicate bacteria, including *S. epidermidis,* within biofilm (9). To date, most of the anti-staphylococcal phages isolated have been characterized in *S. aureus* strains, while only a limited number of studies investigated phage activity against *S. epidermidis* strains (9–12). This is likely a consequence of the particularly laborious approaches needed to isolate phages against *S. epidermidis* (10, 13), but also of their important host range specificity within *S. epidermidis* or their lysogenic properties limiting their therapeutic potential (10, 12). Furthermore, as the scarce anti-*S. epidermidis* phages belong to very different taxonomic genera and families compared to those active against *S. aureus*, and as phages considered for phage therapy are required to be well characterized to ensure the safety of therapeutic products, additional studies to examine all the parameters may be required (production protocols, stability, PK/PD characteristics, immunization, etc.) (14). In this context, phage training could be an interesting alternative to classical phage isolation to expand the host range of previously characterized phages (15). It consists of *in vitro*-directed evolution of phages to force their natural capacity to adapt to resistant bacteria (resistant species herein) with the selection of advantageous mutations and/or recombination events; among the published methods, the Appelmans protocol seems particularly adapted as it relies on the serial expositions of mixtures of phages against resistant bacteria (15).

In the present study, we assessed the potential of phage training to enlarge the host spectrum of anti-*S. aureus* phages already available in our phage bank against *S. epidermidis* clinical isolates. We independently trained two types of mixtures containing high therapeutic potential anti-*S. aureus* phages belonging to two genera, i.e., *Kayvirus* or *Silviavirus* phages, against a selection of seven diverse *S. epidermidis* strains. We showed that, in the conditions used, only *Kayvirus* phages were able to adapt to *S. epidermidis* based on the protocol used and that inter- and intra- sequence types host range specificity was observed.

## RESULTS

### Phage training allows expansion of host range of *Kayvirus* but not *Silviavirus* phages to *S. epidermidis* and is associated with minor reduction of activity against other staphylococcal species

Two mixtures of lytic phages belonging to two genera of the *Herelleviridae* family (myovirus morphotype), either three *Kayvirus* (V1SA09, V1SA12, and V1SA15) or three *Silviavirus* (V1SA19, V1SA20, and V1SA22) phages (Table S1), were trained independently against eight bacterial strains, including seven *S. epidermidis* strains (SE-A to SE-G, Table S2) representative of different sequence types associated with nosocomial infections

including the prevailing ST2 and ST5 (6, 16), and the strain used for phage production (to maintain phage titers along training passages). The ancestral phages had very limited or no activity against the selected *S. epidermidis* strains: phage V1SA09 was the only one active against one of the strains, SE-C with an Efficiency Of Plating (EOP) score of 0.04 (Table 1). After 30 iterative passages, we observed host range expansion only for *Kayvirus* phages. The observed gain of activity for the *Kayvirus* phages mixture occurred only for two *S. epidermidis* bacterial strains and at different time points along phage training, namely after passages 10 (P10) and 30 (P30) for SE-E and SE-G, respectively (Table 1). Of note, phage activity of the mixture against strain SE-E increased dramatically between P10 and P30 with EOP boosted from 0.003 to 7.

Isolation of 20 phage sub-populations was performed using the four bacterial strains for which lytic activity was observed with the P30 phage mixture, namely, SC-Prod-A, SE-C, SE-E, and SE-G (Table 1). Five plaque-forming units (PFU) were isolated on each strain, and named A1 to A5 for those isolated on strain SC-Prod A, C1 to C5 for strain isolated on strain SE-C, etc. All five subpopulations isolated on a given strain had similar phenotypes in terms of lytic activity. The 10 subpopulations isolated on strains SC-Prod-A and SE-C (hereinafter referred to as A1-5 and C1-5 groups, respectively) had the same activity as the ancestral phage V1SA09 (no expansion of host range activity), while the 10 subpopulations isolated on strains SE-E and SE-G (named E1 to E5 and G1 to G5 hereinafter referred to as E1-5 and E1-5 groups, respectively) had extended activity against their respective isolation strain SE-E or SE-G in addition to the initial activity against strains SC-Prod-A and SE-C (Table 1).

The impact of phage training on activity against strains to which phages had never been exposed was also assessed by testing a set of clinical isolates belonging to *S. epidermidis* of various STs (n = 73) and a wider panel of *Staphylococcus* species (*S. aureus*, n = 30; *S. capitis*, n = 10; *S. caprae*, n = 9, *S. haemolyticus*, n = 7; *S. lugdunensis*, n = 9). Again, all five phage sub-populations isolated on a given strain had similar phenotypes. First, for *S. epidermidis* strains, we observed host range expansion only for groups of subpopulations E1-5 and G1-5 with activity against some strains belonging to the same ST as their isolation strains (SE-E and SE-G; Table 2; Fig. 1). Indeed, they were active against 15/47 ST2 isolates and 3/4 ST35 isolates, respectively (Table 2). Conversely, phage sub-populations A1-5 and C1-5 exhibited the same activity spectrum as ancestral phage V1SA09 without any expansion of host range. To further assess the impact of phage training on phage lytic activity, the trained phage E1 was selected as a representative of the group of phage subpopulations E1-5 for the measurement of growth inhibition (liquid assay score [LAS]) of ST2 strains in liquid medium over 24 h (multiplicity of infection [MOI = 1]) in comparison to ancestral phages (Fig. S1). For the 15 ST2 strains against which phage E1 exhibited an EOP score >0, the phage was also active in liquid medium with a mean LAS of 89%. Trained phage E1 showed activity in liquid medium against one additional strain while displaying an EOP score of 0. Finally, we observed no

**TABLE 1** Evolution of activity of phages against *S. epidermidis* strains during training compared to ancestral phages measured with the spot test[b]

| Bacterial strain | Ancestral phages | | | Phage training mixtures | | | | | | Phage subpopulations isolated from P30 | | | |
|---|---|---|---|---|---|---|---|---|---|---|---|---|---|
| | V1SA09 | V1SA12 | V1SA15 | P5 | P10 | P15 | P20 | P25 | P30 | A1-5[a] | C1-5[a] | E1-5[a] | G1-5[a] |
| SC-Prod-A | 1 | 1 | 1 | 1 | 1 | 1 | 1 | 1 | 1 | 1 | 1 | 0.1–0.3 | 0.1–0.6 |
| SE-B | 0 | 0 | 0 | 0 | 0 | 0 | 0 | 0 | 0 | 0 | 0 | 0 | 0 |
| SE-C | 0.04 | 0 | 0 | 0.02 | 0.05 | 0.09 | 0.05 | 0.03 | 0.003 | 0.05–0.2 | 0.06–0.5 | 0.03–0.1 | 0.06–0.3 |
| SE-D | 0 | 0 | 0 | 0 | 0 | 0 | 0 | 0 | 0 | 0 | 0 | 0 | 0 |
| SE-E | 0 | 0 | 0 | 0 | 0.003 | 0.5 | 0.1 | 0.5 | 7 | 0 | 0 | 1 | 0 |
| SE-F | 0 | 0 | 0 | 0 | 0 | 0 | 0 | 0 | 0 | 0 | 0 | 0 | 0 |
| SE-G | 0 | 0 | 0 | 0 | 0 | 0 | 0 | 0 | 0.3 | 0 | 0 | 0 | 1 |
| SE-H | 0 | 0 | 0 | 0 | 0 | 0 | 0 | 0 | 0 | 0 | 0 | 0 | 0 |

[a]All five phage subpopulations isolated either on strains A, C, E, or G exhibited similar host range.
[b]Reference strains used for efficiency of plating (EOP) calculation were SC-Prod-A (strain used for ancestral phages amplification) for phage training mixtures and respective isolation strains for isolated subpopulations. Mean EOP values obtained from experiments performed in triplicate are indicated. For phage subpopulations isolated from P30 phage training mixture, range of values obtained for the five subpopulations isolated on each bacterial strain are indicated for simplification of the table.

**TABLE 2** Comparison of activity of ancestral and trained phages against a large panel of strains belonging to six *Staphylococcus* species measured using the spot test[e]

| Bacterial species | Ancestral phages | | | Phage subpopulations isolated from P30 mixture | | | |
|---|---|---|---|---|---|---|---|
| | V1SA09 | V1SA12 | V1SA15 | A1-5[a] | C1-5[a] | E1-5[a] | G1-5[a] |
| *S. epidermidis* | | | | | | | |
| ST2 (*n* = 47)[b] | 1 | 0 | 0 | 1 | 1 | 15 | 1 |
| ST35 (*n* = 4)[c] | 0 | 0 | 0 | 0 | 0 | 0 | 3 |
| Other (*n* = 22) | 4[d] | 0 | 0 | 4[d] | 4[d] | 4[d] | 4[d] |
| *S. aureus* (*n* = 30) | 14 | 16 | 4 | 14 | 14 | 3 | 13 |
| *S. capitis* (*n* = 10) | 10 | 9 | 10 | 10 | 10 | 10 | 7 |
| *S. caprae* (*n* = 9) | 2 | 0 | 0 | 2 | 2 | 0 | 0 |
| *S. haemolyticus* (*n* = 7) | 6 | 5 | 2 | 6 | 6 | 2 | 5 |
| *S. lugdunensis* (*n* = 9) | 6 | 5 | 4 | 6 | 6 | 2 | 5 |

[a]All five phage subpopulations isolated either on strains A, C, E, or G exhibited similar host range.
[b]Strain SE-E used for the isolation of subpopulations E1 to E5 belonged to ST2.
[c]Strain SE-G used for the isolation of subpopulations G1 to G5 belonged to ST35.
[d]All strains susceptible to phages belonged to ST5.
[e]Numbers of strains for which phages exhibited an EOP score > 0 are indicated.

impact of training using *S. epidermidis* strains on the activity against other staphylococcal species, and even a loss of activity of trained phage subpopulations against few strains compared to the ancestral phage with the most narrow host spectrum. For instance, trained phage subpopulations E1-5 were active against 2 out of 9 *S. lugdunensis* strains or 3 out of 30 *S. aureus* strains while ancestral phages were active at least against four out of nine or four out of thirty of these same strains (Table 2).

## Phage training is associated with the recombination of genomes of ancestral phages

Genomes of trained phage sub-populations were sequenced and compared to those of ancestral phages. Consistently with host range assessment results showing no change in host range for sub-populations A1-5 and C1-5, their genomes were identical to that of V1SA09 (100% coverage and identity). Conversely, coverage and identity between genomes of sub-populations E1-5 or G1-5 and ancestral phages varied from 85% to 94% and from 96.2% and 99.5%, respectively (Table 3). Of note, three and two variants of genomes were, respectively, identified in the two groups of sub-populations E1-5 or G1-5, exhibiting the same host range within each group but different host range in between the two groups. Among subpopulations E1-5, genomes of E1, E3, and E5 were identical but different from genomes of E2 and E4, the latter being different from each other. Among subpopulations G1-5, genomes of G1 and G3 and those of G2, G4, and G5 were identical. These genomes were found to have evolved from those of ancestral phages V1SA09 and V1SA15 with at least 10 to 15 recombination events (Table 3; Table S3; Fig. S2). A single small 150 bp region was predicted to be inherited from V1SA12 in phage subpopulations E1, E3, E4, and E5 with 100% identity inside a larger region inherited from V1SA15. However, the former could also have originated from V1SA15 with an excess accumulation of SNPs in this region. Predicted recombination events occurred mainly in two regions of the genome: the first comprising numerous genes coding for hypothetical proteins and genes involved in DNA replication, and the second comprising structural genes (Fig. S2). Most of these recombination events had no impact on the sequence of genes near the recombination sites which was identical to that of a gene of at least one ancestral phage. Genes with unmodified sequences compared to ancestral genes were unlikely related to the modifications of host range. Thus, we focused our analyses on genes showing mutations compared to the two ancestral phages (7 to 13 genes per trained phage genome). Six to nine of these mutated genes were associated with amino acid changes in the corresponding proteins (Table 4). Genes that were identical for each of the phage subpopulations of group E (same host range), but which were different from the genes of the phage subpopulations of group G

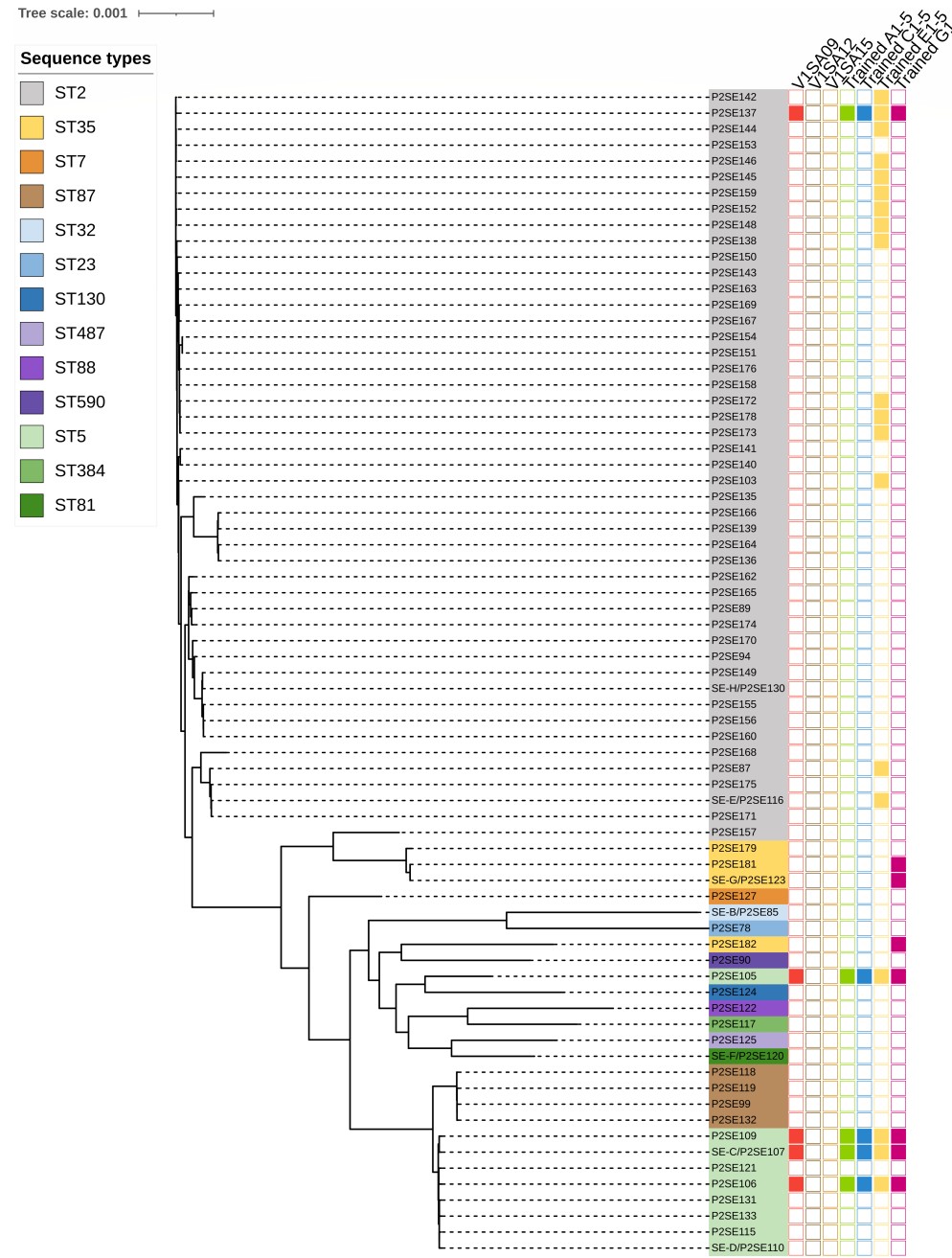

**FIG 1** Activity of ancestral and trained phages against a large and diverse collection of *S. epidermidis* isolates. Activity of phages against strains of various sequence types (indicated by the color used on strain names) is shown in columns on the right part of the phylogenetic tree (activity/no activity: full/empty square).

(different host range compared to group E)—and vice versa—were of particular interest. Indeed, they indicate convergence and are, therefore, likely related to the extended host range of these trained phage subpopulations. We identified two (Table 4, lines 6 and 7) and four (Table 4, lines 1–3 and 10) genes corresponding to this definition in groups E1-5 and G1-5, respectively. Unfortunately, they mainly corresponded to hypothetical proteins precluding any conclusion as to a causal effect. Finally, two mutated genes were identical in all trained phage subpopulations (Table 4, lines 12–13) and were, thus, likely not related to the different evolved phenotypes of groups E1-5 and G1-5.

**TABLE 3** Analysis of genomes of trained phage subpopulations E1-5 and G1-5 compared to those of ancestral phages

| Phage sub-popula-tions | Accession number | Size (pb) | Coverage/identity of ancestral genomes (%) | | | Recombination events (n) | Origin of genes in trained phages | |
|---|---|---|---|---|---|---|---|---|
| | | | V1SA09 (Cov/Id) | V1SA12 | V1SA15 | | V1SA09 (%) | V1SA15 (%) |
| E1/E3/E5 | OR640346 | 142.874 | 88/99.5 | 85/96.5 | 94/99.4 | 11 | 51.6[a] | 48.3[a] |
| E2 | OR640347 | 142.797 | 90/99.3 | 87/97.0 | 92/99.4 | 12 | 62.4 | 37.6 |
| E4 | OR640348 | 143.293 | 88/98.6 | 85/96.2 | 94/99.4 | 15 | 52.7[a] | 47.2[a] |
| G1/G3 | OR640349 | 140.937 | 93/99.4 | 88/97.2 | 90/98.3 | 10 | 69.2 | 30.8 |
| G2/G4/G5 | OR640350 | 141.497 | 93/98.8 | 87/96.2 | 90/99.4 | 10 | 61.3 | 93.1 |

[a]A 125 bp portion of a gene located in a region likely inherited from V1SA15 was considered to be inherited of a recombination with V1SA12 genome.

## Host range expansion to ST2 strains is mediated by the improvement of a step of the viral cycle after adsorption to bacteria

Further analyses were focused on the activity of the trained phage E1 on ST2 strains as a representative of the group of subpopulations E1-5 with the same phenotype isolated on strain SE-E. Adsorption of this phage onto bacterial strain SE-E was not modified compared to that of ancestral phage V1SA09 whatever time points (Fig. 2): less than 5% of free phages were detected in the supernatant after 24 min for both phages. Conversely, adsorption of this same trained phage was higher than that of ancestral phage V1SA15 for which a minimum of 10% of free phages was measured after 40 min. In addition, the concentration of free phages rose dramatically after 40 min only for phage E1 indicating that it was the only one able to multiply in this bacterial strain. Interestingly, phage V1SA19, used as representative of *Silviavirus* phages and negative control of phage activity and training, did not adsorb onto this bacterial strain.

Since the host range expansion of this trained phage subpopulation E1 compared to the ancestral phages was limited to ST2 strains, it was hypothesized that the latter may harbor specific genes related to a phage resistance mechanism limiting the activity of the ancestral phages and overcome by the trained phages. Thus, we searched for specific genes of this ST among the set of *S. epidermidis* strains tested in the present study using a pangenome approach. Eight candidate genes were identified in 46 of the 47 ST2 strains. Of note, even if the remaining strain, which was resistant to the trained phages, belonged to ST2, it was phylogenetically distant from all other ST2 strains (P2SE157; Fig. 1). These eight genes were clustered in bacterial genomes and were all located in a region of 41 genes or less corresponding to a prophage sequence showing 90%–95% sequence identity with PI-Sepi-HH2 prophage previously reported to be associated with ST2 strains virulence (Fig. S3) (6). They were annotated or predicted as hypothetical proteins, helix-turn-helix transcriptional regulators, phage tail tape measure protein, excisionase, or peptidoglycan recognition family protein (Table S4). Interestingly, one of these proteins was predicted as a "SaV-like" protein based on the analysis of protein domains (using InterPro software), which has been previously described as involved in susceptibility to abortive infection systems halting phage propagation in *Lactococcus* strains (17).

A genome-wide association study (GWAS with DBGWAS tool) was also conducted to explore the mechanisms associated with the resistance to trained phages (31/47 ST2 strains) (18). With this approach, 293 significant nodes were identified. Most of them (75%) were associated with phage resistance. However, none of them corresponded to sequences of genes previously reported as involved in resistance to phages (PADLOC and DefenseFiner databases [19, 20]). Interestingly, a succession of nodes corresponding to a group of five genes was identified in 30/31 resistant strains and in only 2/16 susceptible ones. The predicted functions of these genes included a recombinase, a *S. aureus* uracil-DNA glycosylase inhibitor (SAUGI) family, and three hypothetical proteins. To assess the possible role of this cluster of genes in resistance to trained phages, this group of five genes was cloned into the expression vector pSK265 and we transformed

**TABLE 4** Genes of trained phages showing nucleotide mutations associated with amino acid changes in the corresponding proteins compared to ancestral phages

| Annotation (example phage_gene number) | E1/E3/E5 | | E2 | | E4 | | G1/G2 | | G2/G4/G5 | |
|---|---|---|---|---|---|---|---|---|---|---|
| | V1SA9 | V1SA15 | V1SA9 | V1SA15 | V1SA9 | V1SA15 | V1SA9 | V1SA15 | V1SA9 | V1SA15 |
| **Region 1** | | | | | | | | | | |
| 1  Phage protein (G1_1) | / | / | / | / | / | / | **95.2** | **99.9** | **95.2** | **99.9** |
| 2  Phage protein (G1_9) | / | / | / | / | / | / | **97.6** | **98.1** | **97.6** | **98.1** |
| 3  Phage protein (G1_15) | 98.1 | 91.2 | 98.9 | / | 98.1 | 91.2 | **90.6** | **98.8** | **90.6** | **98.8** |
| 4  Phage protein (E2_15) | / | / | / | 91.1 | / | / | / | / | / | / |
| 5  Hypothetical protein (E1_24) | 91.5 | 27.0 (78) | / | / | 91.5 | 27.0 (78) | NH | 98.5 | NH | 98.5 |
| 6  Hypothetical protein (E1_56) | NA | **97.6 (95)** | NA | **97.6 (95)** | NA | **97.6 (95)** | NH | NH | NH | NH |
| 7  Hypothetical protein (E1_61) | **93.9** | **92.7** | **93.9** | **92.7** | **93.9** | **92.7** | / | / | / | / |
| **Region 2** | | | | | | | | | | |
| 8  Recombination-related exonuclease (E4_154) | / | / | / | / | 99.5 | 99.7 | / | / | / | / |
| 9  Carbohydrate binding domain-containing protein[b] (E2_163) | / | / | 99.8 | 99.2 | 99.8 | 99.2 | NH | NH | NH | NH |
| 10  Tail fiber protein[b] (G1_158) | / | / | / | / | / | / | **99.9** | **99.9** | **99.9** | **99.9** |
| 11  Tail fiber protein[b] (E1_165) | 99.8 | 98.6 | / | / | / | / | / | 98.5 | 98.5 | 99.9 |
| 12  Phage tail sheath (E1_186) | 99.5 | 99.8 | 99.5 | 99.8 | 99.5 | 99.8 | 99.5 | 99.8 | 99.5 | 99.8 |
| 13  Phage holing (E1_222) | 99.4 | 98.8 | 99.4 | 98.8 | 99.4 | 98.8 | 99.4 | 98.8 | 99.4 | 98.8 |

Header spanning note: **Protein identity between trained and ancestral phages (coverage)[a]**

[a]Coverage of the ancestral protein sequence is indicated if <100%; /: protein sequence identical to one ancestral gene; NH, no hit identified. Genes common within groups of sub-population E1-5 or G1-5 exhibiting the same host range within each grouop but different host range between the two groups are in bold.

[b]Annotation was improved by running NCBI blastp.

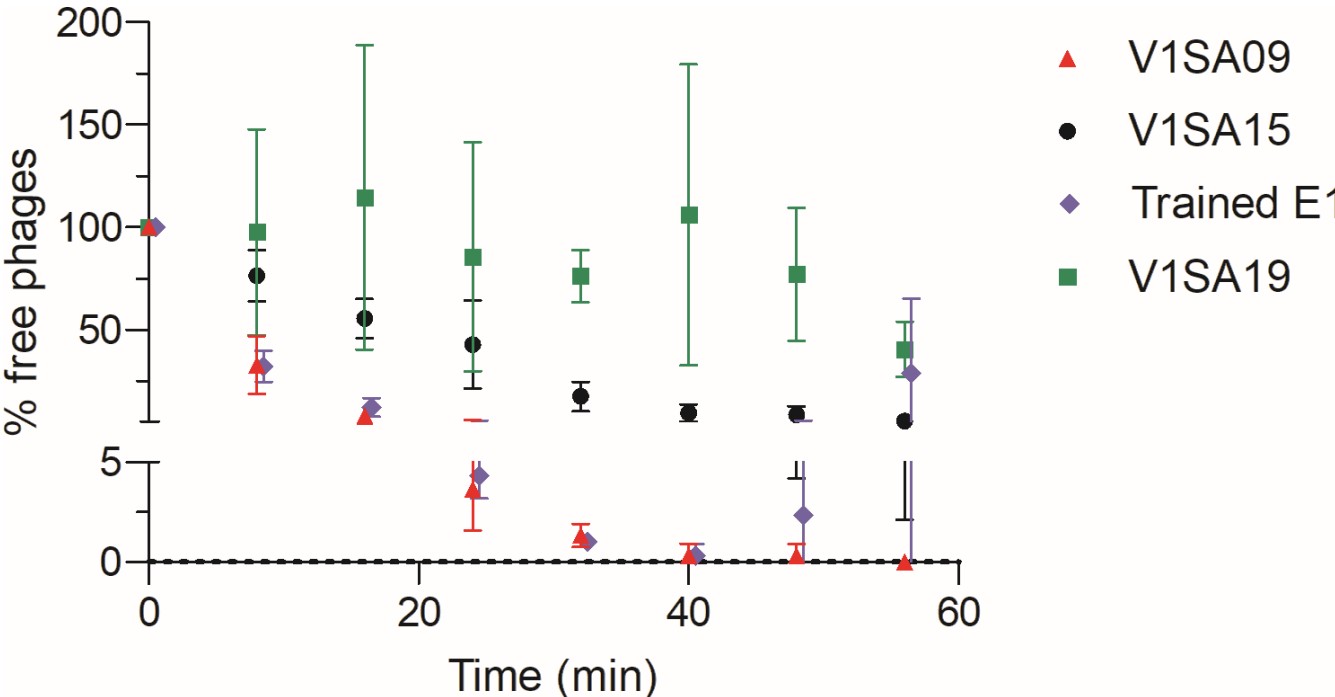

**FIG 2** Adsorption of ancestral and trained phages onto SE-E *S. epidermidis* ST2 strain. Means ± standard deviations are indicated. Phage V1SA19 was used as a representative of *Silviavirus* phages and as a comparator for the adsorption of ancestral V1SA09 and V1SA15 phages, and trained E1 *Kayvirus* phages.

the P2SE87 strain, which was susceptible to the trained phage E1. However, complementation had no impact on phage susceptibility of this strain despite effective plasmid-mediated expression of the SAUGI-coding gene assessed using qRT-PCR (a 107-fold superior to expression by *S. epidermidis* strain P2SE175 which was resistant to the phage).

## DISCUSSION

Phage training by the principles of the Appelmans protocol has been used to extend the host range of phages infecting several bacterial species, including *S. aureus* (15, 21). In the present study, we evaluated whether this protocol could be applied to extend the host range of anti-*S. aureus Kayvirus* and *Silviavirus* phages to *S. epidermidis,* a major human pathogen for which alternatives to antibiotics are urgently needed, and for which the environmental isolation and characterization of active phages is laborious and time-consuming.

In the conditions used in the present study, only *Kayvirus* but not *Silviavirus* phages were successfully trained with the improved activity of the final training mixture against two of the seven selected *S. epidermidis* strains compared to the ancestral phages. Interestingly, the data presented herein indicate that only ancestral *Kayvirus* but not *Silviavirus* phages were able to adsorb to one of these strains, which may explain why *Silviavirus* could not be adapted to *S. epidermidis* isolates. In addition, *Kayvirus* phages are known to exhibit a polyvalent activity targeting several *Staphylococcus* species, suggesting that their ability to adapt to a different species is superior to that of other *S. aureus*-specific phages such as *Silviavirus* (22). Indeed, previous studies have shown that the two different phage genera recognize different bacterial receptors. *Kayvirus* phages can adsorb to sugar moieties and/or the backbone of wall teichoic acids (WTA), while *Silviavirus* only adsorb to the latter (23–25). WTA composition varies between *S. aureus* and coagulase-negative staphylococci as the former species produces WTA with on a phospho-ribitol polymer backbone attached to two units of glycerol, whereas the latter produces phospho-glycerol WTA (26, 27). Thus, *Kayvirus* phages are likely able to adsorb

to a pattern of the WTA conserved between *Staphylococcus* species accounting for their polyvalent activity.

Isolation of phage subpopulations from the final training mixture on four different bacterial strains showed that only those isolated on the strains against which we observed improved activity after training had a wider host range compared to the ancestral phages. Conversely, subpopulations isolated on the strains against which one of the ancestral V1SA09 phage was active turned out to be identical to this phage, suggesting that it remained the major population active against these strains after 30 co-culture passages. Phage subpopulations with extended host range had two types of phenotypes, being active on their isolation strain but also on strains of the same sequence types (ST2 or ST35). Sáez Moreno et al. previously reported the use of phage training to improve the activity of different genera of phages against *S. aureus* isolates, including isolates which had never been in contact with the phages (21). However, the significance of the host range changes reported by these authors is difficult to interpret as they did not analyze phage activity according to *S. aureus* sequence types. In addition, we showed that phage adaptation to *S. epidermidis* was counterbalanced by a minor decrease of activity against other *Staphylococcus* species. This phage specialization could be expected as it has been reported in previous studies, notably in natural environments where phage adaptation to "sub-optimal" hosts is a driver for phage diversification to survive and maintain viral populations (28). Our study was limited to the training of phages isolated on *S. aureus* strains and with no or little activity against *S. epidermidis* with the aim of limiting the number of phage genera in our collection. However, given the important specificity of the trained phages obtained, it would be interesting to compare the efficacy of training specific anti-*S. epidermidis* phages with our results.

The analysis of trained phage genomes indicated that they were mosaics of their ancestors used in the training mixture. Mosaicism is the main natural evolutionary pathway of phages and includes two types of recombination events, either between large homologous regions whose extremities are specifically recognized by recombinases, or randomly between short sequences (29). Such mosaicism has also been previously reported in studies of *in vitro* training of phage mixtures (15, 21). Based on the large panel of evolved phages sequenced in the present study, we demonstrate that adapted phages isolated on a given bacterial strain can present the same phenotype while exhibiting a different genomic mosaicism. We compared the genomes of evolved phages to those of ancestral ones to identify genes with evolved sequences compared to both ancestral phages that could be associated with host range expansion. Only two and four mutated genes were common to all phage subpopulations in each group with different phenotypes and would, thus, merit further exploration. Alternatively, the expansion of host range might also be due to the accumulation of multiple interdependent mutations affecting different genes rather than to single events. Most of these viral genes were located far from the structural module involved in bacterial host recognition. This suggests that phage adaptation is likely not related to changes in adsorption to the bacterial surface but to subsequent steps of phage infection. This hypothesis is supported by the data of the adsorption assays showing no increase in adsorption for evolved phages compared to ancestral ones. This is consistent with findings of other studies describing that host range adaptation of *Kayvirus* phages was associated with mutations in genes in the genome module allowing host machinery to takeover without change in adsorption (21, 30, 31). However, this contrasts with the well-described evolutionary pathways of phage active against Gram negative bacteria, such as *Escherichia coli*, which adapt to resistant bacterial strains by recognition of a secondary receptor (32).

Adapted phages selected in the present study had important inter- and intra-sequence type specificity, having activity against strains belonging to only ST2 or ST35. Narrow host spectrum has previously been reported for other types of anti-*S. epidermidis* phages (10, 12). This could be related to the multiple determinants of phage activity along the viral replication cycle that remain poorly studied in *S. epidermidis*. We further

attempted to characterize bacterial genetic determinants of the activity of trained phages among ST2 strains which is of particular interest due do its high prevalence compared to other sequence types (6, 7, 33). We identified a cluster of genes specific of ST2 strains and indicating the presence of a prophage related to the prophage PI-Sepi-HH2, previously associated with virulence in ST2 strains (6). This prophage may harbor a resistance mechanism overcome by trained phages. Of note, Kuntova et al. also reported that a prophage-encoded protein provided resistance to *Kayvirus* phages in *Staphylococcus aureus* causing abortive infection and that the phage could overcome this resistance after adaptive laboratory evolution (30). Further studies, that are beyond the scope of the present work, are required to assess whether one or several of the genes carried by the ST2-specific prophages identified in the present study could impair the replication of *Kayvirus* phages and whether one or more mutations in trained phages could counteract this resistance mechanism. Of note, some of these eight genes are less likely to be involved in the resistance mechanism such as those encoding the prophage tail-tape measure protein or the excisionase related to the prophage replication. In addition, the data presented herein indicate that only a subset of ST2 strains were susceptible to trained phages. The presence of five genes, identified using a GWAS approach, including one annotated as coding for a uracil-DNA glycosylase inhibitor (SAUGI) was significantly associated with phage resistance. The SAUGI inhibits the elimination of uracils in DNA that result from the spontaneous deamination of cytosine or the incorporation of dUTP by mistake during replication (34). As the replication of viruses can be altered by the incorporation of dUTP, its role in phage resistance has previously been suggested (34). *Kayvirus* phages are notably likely susceptible to genomic uracilation as they encode dUTPase (35). By preventing the removal of uracil from phage DNA, SAUGI may, thus, act as an anti-phage factor. However, we were not able to demonstrate this as the plasmid-mediated expression of these five genes in a *S. epidermidis* strain susceptible to trained phages did not impact phage activity. Deletion of this region in a phage-resistant strain could be an alternative to plasmid complementation. However, genetic manipulation of *S. epidermidis* strains is particularly challenging notably because of the multi-drug resistance of ST2 strains, including macrolides and chloramphenicol, limiting the choice of plasmids that can be used for such experiments.

To conclude, we show that phage training can be used to extend the host range of polyvalent *Kayvirus* phages within the *Staphylococcus* genera to the *S. epidermidis* species. In the context of the rapid development of phage therapy, the adaptation of previously characterized phages may be an attractive alternative to the fastidious repeated isolation of new phages to improve the therapeutic potential of a phage collection.

## MATERIALS AND METHODS

### Bacterial and phage strains

All bacterial strains used in the present study are listed in Table S2 (*n* = 138 strains including 73 *S. epidermidis* strains and 65 strains of other *Staphylococcus* species). They were obtained from the collections of the French National Reference Center for Staphylococci and the bacteriology department of the Lyon University Hospital (France). They were cultured from frozen stocks onto blood agar plates (bioMérieux, Marcy-l'Etoile, France) at 37°C. Overnight liquid cultures were then obtained from individual colonies incubated in tryptic soy broth (TSB; bioMérieux) under agitation at 37°C.

Phages included in this study belonged to two different genera, namely *Kayvirus* (vB_SauM-V1SA09, vB_SauM-V1SA12, and vB_SauM-V1SA15, here designated V1SA09, V1SA12, and V1SA15, respectively, for simplification) or *Silviavirus* (vB_SauM-V1SA19, vB_SauM-V1SA20, and vB_SauM-V1SA22, designated V1SA19, V1SA20, and V1SA22, respectively) phages, and were respectively isolated for the purpose of this study (Table S1) or previously described by our team (36). They were propagated using low-virulent *S.*

*aureus* or *S. capitis* strains and, if possible, which did not contain any prophages in their genome as previously reported (36).

## Phage training (Appelmans protocol) and isolation of phage subpopulations from training mixture

The Appelmans protocol was carried out as previously reported by Burrowes et al. (15). Briefly, three phages were combined in a 1:1:1 mixture ($1 \cdot 10^{10}$ PFU/mL of each phage). In a 96-well microtiter plate, 90 µL of this mixture (dilution $10^0$) and of serial 10-fold dilutions ($10^{-1}$ to $10^{-9}$) were added in each of the 10 first columns (one dilution/column). The two last columns contained only TSB medium. Then, 10 µL of overnight bacterial liquid culture was added to wells 1–11 of each of the eight lines (one different bacterial strain/line). Wells no. 11 and no. 12 were, thus, used as positive and negative growth controls, respectively. After overnight incubation at 37°C under 180 rpm agitation, wells showing complete or partial lysis compared to the positive control well, as well as the first completely turbid well, were harvested and pooled. This mixture was then filtrated using a 0.22-µm syringe filter to remove living and dead bacteria; this lysate was called the training mixture of passage 1 (P1). It was then used to inoculate a new plate as described above using new overnight cultures of the same bacterial strains to perform passage 2 (P2). These steps were repeated up to passage 30 (P30). All training mixtures were stored at 4°C. Five individual PFUs obtained from P30 mixture in a 10-mL TSB soft agar layer containing 250 µL of one of the four selected bacterial strains were isolated performing five rounds of purification on each bacterial strain. In total, 20 phage subpopulations were isolated using four bacterial strains (SC-Prod-A, SE-C, SE-E, SE-G). They were then propagated using their respective isolation strain in TSB medium with a MOI of $10^{-2}$.

## Phage host range assessment

Phage activity was assessed onto all bacterial strains using the spot test assay and EOP score determination as previously described (36). A bacterial strain was considered susceptible to the phage if individual PFUs could be observed at any dilution (EOP > 0). In addition, for the selection of ST2 *S. epidermidis* strains, we further characterized phage activity in a liquid kinetic assay measuring bacterial growth inhibition using spectrometry (optical density at 600 nm [$OD_{600}$]) in the presence of phages at MOI = 1 compared to untreated bacteria and calculation of the LAS score (37). Assays were performed with three biological replicates.

## Adsorption assays

The adsorption assays were performed as previously described (12). Briefly, selected *S. epidermidis* strains were grown in 10 mL TSB at 37°C under agitation (180 rpm) to reach an $OD_{600}$ of 0.4 corresponding to a bacterial concentration of approximately $10^8$ CFU/mL. Phages were then added at a MOI of $10^{-2}$ ($10^6$ PFU/mL) and incubated at 37°C. Aliquots of 300 µL were taken every 8 min for 60 min and 0.22 µm was filtered to collect non-adsorbed phages. Serial dilutions were performed and spotted onto a soft agar containing the production strain to determine the titer of non-adsorbed phage particles. Assays were formed with three biological replicates and three replicates of phage titration. Percentages of free phages were compared using the Mann-Whitney test considering a *P*-value ≤ 0.05 as significant.

## Bacterial genome sequencing and analysis

Whole genome sequencing was performed using Illumina (San Diego, CA, USA) short-read technology (Nextera XT or DNA prep kit for library preparations and NextSeq sequencer) with a read length of $2 \times 150$ bp. Genomes were assembled using SPAdes. Multi-locus sequence typing (MLST) was performed using the PubMLST database (https://pubmlst.org). Additionally, one of the ST2 strains (P2SE160) was sequenced using

Oxford Nanopore (Oxford, UK) long-read technology. Hybrid-like assemblies of other ST2 genomes were produced by mapping contigs assembled from Illumina sequencing against P2SE160 genome (ragtag scaffold v2.1.0 [38]). Bacterial genomes were annotated using Bakta v1.7.0 (39). A GWAS was performed to compare phage-susceptible and-resistant ST2 strains using DBGAS v0.5.4 (18). We analyzed whether significant nodes identified corresponded to sequences of genes annotated by the two anti-phage systems databases DefenseFinder and PADLOC (19, 20) or prophage sequences detected by PHASTER (40). After pan-genome analysis using Roary (41), ST2-specific genes were identified. InterPro and HHpred tools were used to assign their function and identify protein domains in addition to gene annotation (42, 43). Core genome genes obtained with Roary were used to construct a phylogenetic tree with FastTree v2.1.11 (44).

## Bacterial complementation

The plasmid pSK265, a derivative of pC194, was used to express the five genes identified by the GWAS under the control of the *rpoB* promoter (45). Gibson assembly (New England Biolabs, Ipswich, MA, USA) was used for the construction of the plasmid with the following primers: psk265_F: GACACTACA CCACCACTAAGATCCTAAAC; psk265_R: GATTG ATCATCCCATCTCACCCC; insert_F: GTGAGATGGGATGATCAATCAACATGAAGCG, insert_R: A GTGGTGGTGTAGTGTCATGGTTCATAGA. The Gibson assembly product was first electroporated into *S. aureus* RN4220 strain using 2 µL of the Gibson assembly product and electroporation conditions previously described (46). Electroporated cells were grown onto TSA plates supplemented with chloramphenicol at 20 µg/mL. The presence of plasmids with the correct insert in the resulting recombinant colonies was confirmed by PCR with primers overlapping the insert and the plasmid: control_plasmid: ATAAGAGCA GGGAAAGAAACCGT; control_insert: TTCCTCATTGAGTCCTTTTC GTCTT. The plasmid was then amplified and extracted from a single colony (Plasmidi Midiprep kit; ZymoResearch, Irvine, CA, USA) to allow the transformation of 1 µg into *S. epidermidis* P2SE87 strain. The plasmid sequence was verified by Illumina sequencing. The expression of the SAUGI-coding gene on the plasmid insert and of the *gyrB* housekeeping gene was assessed after RNA extraction (RNeasy Plus Mini kit; Qiagen, Hilden, Germany) from overnight cultures in triplicates followed by reverse transcription (Reverse Transcription System; Promega, Madison, WI, USA) and qPCR (TB green assay, Takara, Kyoto, Japan and QuantStudio thermocycler, ThermoFisher, Waltham, MA, USA) with the following primers: SAUGI_F: AATGTGAAGCACTTGAGGAAGCAGC; SAUGI_ R: TCTGTGAGGT ATAGAAAGTCCATATCAAAA TGATC; gyrB_F: CGGTTCGTAAAAGACCGGGTA; gyrB_R: CAGGGCGTCCCATC TTTTCT. Fold change expression was assessed with the $2^{-\Delta\Delta Ct}$ method.

## Phage genome sequencing and analysis

Phage DNA extraction, sequencing, and assembly were performed as previously described (36). Genomes were first annotated using BV-BRC v3.30.5i with parameters for phage annotation (47). Other databases were successively used for genes annotated as "hypothetical protein" or "phage protein," in order of use: PhaLP, pVOGs, PHROG, and Phantome databases. Prediction of protein domain by InterPro was also performed to improve the prediction of function of hypothetical proteins of interest. Genomes were deposited in Genbank under the accession numbers indicated in Table S1; Table 3. Because of the important homology between ancestral phages and the necessity of performing multiple pairwise comparisons, alignments of whole-genome sequences inaccurately reflected the recombination events. To overcome this problem, to compare trained phages with ancestral phages, each gene of trained phages was blasted against ancestral genes to compare sequence coverage and identity to infer their origin. In cases where a gene in a trained phage had 100% identity with only one ancestral phage, the gene was considered to have been inherited from the latter. In cases where a gene from a trained phage had 100% identity with two ancestral phages, the origin of the surrounding genes was considered (Table S3). Single-nucleotide polymorphisms were

identified using Snippy v4.6.0 after mapping of trained phage reads against ancestral phage-assembled genomes to confirm mutations found with the blast approach.

## ACKNOWLEDGMENTS

The authors acknowledge Leslie Blazere and Emilie Helluin for their technical support. The authors would like to thank the technicians, engineers, and biologists of the French National Reference Centre for Staphylococci for their help. Sequencing of phage genomes was performed with the GenePII platform of Hospices Civils de Lyon. The authors also acknowledge the scientific review service of Hospices Civils de Lyon for the English writing proofreading.

This study was supported by the French Agence Nationale pour la Recherche (PHAG-ONE project, ANR 20-PAMR-0009) and the Fondation Hospices Civils de Lyon.

## AUTHOR AFFILIATIONS

[1]Service de bactériologie, Centre National de Référence des Staphylocoques, Institut des Agents Infectieux, Hospices Civils de Lyon, Lyon, France
[2]Equipe StaPath, CIRI, Centre International de Recherche en Infectiologie, Inserm U1111, Université Claude Bernard Lyon 1, CNRS UMR5308, ENS de Lyon, Lyon, France
[3]Bacteria, Parasites and Fungi, Statens Serum Institut, Copenhagen, Denmark
[4]Antimicrobial Resistance and Infectious Diseases Laboratory, Harry Butler Institute, Murdoch University, Perth, Australia
[5]GIMAP Team, CIRI, Centre International de Recherche en Infectiologie, Inserm U1111, Université Claude Bernard Lyon 1, CNRS UMR5308, ENS de Lyon, Lyon, France
[6]Faculty of Medicine, Université Jean Monnet St-Etienne, St-Etienne, France
[7]Department of Infectious Agents and Hygiene, University Hospital of St-Etienne, St-Etienne, France
[8]Faculté de Pharmacie, Université Claude Bernard Lyon 1, Lyon, France

## AUTHOR ORCIDs

Camille Kolenda http://orcid.org/0000-0002-3716-0521
Marc Stegger http://orcid.org/0000-0003-0321-1180

## FUNDING

| Funder | Grant(s) | Author(s) |
| --- | --- | --- |
| Agence Nationale de la Recherche (ANR) | ANR 20-PAMR-0009 | Camille Kolenda |
| | | Mélanie Bonhomme |
| | | Mathieu Medina |
| | | Floriane Laumay |
| | | Frédéric Laurent |
| HCL \| Fondation Hospices Civils de Lyon (Fondation HCL) | | Camille Kolenda |
| | | Mathieu Medina |
| | | Frédéric Laurent |

## AUTHOR CONTRIBUTIONS

Camille Kolenda, Conceptualization, Data curation, Formal analysis, Funding acquisition, Investigation, Methodology, Writing – original draft | Mélanie Bonhomme, Formal analysis, Investigation, Software, Writing – review and editing | Mathieu Medina, Formal analysis, Funding acquisition, Project administration, Writing – review and editing | Mateo Pouilly, Investigation, Writing – review and editing | Clara Rousseau, Investigation, Writing – review and editing | Emma Troesch, Investigation, Writing – review and editing | Patricia Martins-Simoes, Investigation, Writing – review and editing | Marc Stegger,

Investigation, Writing – review and editing | Paul O. Verhoeven, Methodology, Writing – review and editing | Floriane Laumay, Methodology, Supervision, Writing – review and editing | Frédéric Laurent, Funding acquisition, Project administration, Supervision, Writing – review and editing

## DATA AVAILABILITY

Accession numbers for sequencing data are indicated in Table S2 (project accession numbers PRJEB69270 and PRJEB72624).

## ADDITIONAL FILES

The following material is available online.

### Supplemental Material

**Supplemental Figures (mSystems00850-24-s0001.docx).** Figures S1 to S3.
**Table S1 (mSystems00850-24-s0002.docx).** Phage collection.
**Table S2 (mSystems00850-24-s0003.xlsx).** Bacterial strain collection.
**Table S3 (mSystems00850-24-s0004.xlsx).** Prediction of recombination events between phage genomes.
**Table S4 (mSystems00850-24-s0005.docx).** Genes specifically identified in ST2 strains.

### Open Peer Review

**PEER REVIEW HISTORY (review-history.pdf).** An accounting of the reviewer comments and feedback.

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
