## [Reviewer comments · mSystems]

Potential of training of anti-*Staphylococcus aureus* therapeutic phages against *Staphylococcus epidermidis* multidrug-resistant isolates is restricted by inter- and intra-sequence type specificity

Camille KOLENDA, Mélanie Bonhomme, Mathieu Medina, Mateo Pouilly, Clara Rousseau, Emma Troesch, Patrícia Martins Simões, Marc Stegger, Paul Verhoeven, Floriane Laumay, and Frédéric Laurent

Corresponding Author(s): Camille KOLENDA, Hospices Civils de Lyon

Review Timeline:

Submission Date:

June 25, 2024

Accepted:

June 27, 2024

Editor: Michela Gambino

Reviewer(s): The reviewers have opted to remain anonymous.

Transaction Report:

DOI: <https://doi.org/10.1128/msystems.00850-24>

Re: mSystems00850-24 (Potential of training of anti-*Staphylococcus aureus* therapeutic phages against *Staphylococcus epidermidis* multidrug-resistant isolates is restricted by inter- and intra-sequence type specificity)

Dear Dr. Camille KOLENDA:

Your manuscript has been accepted, and I am forwarding it to the ASM production staff for publication. Your paper will first be checked to make sure all elements meet the technical requirements. ASM staff will contact you if anything needs to be revised before copyediting and production can begin. Otherwise, you will be notified when your proofs are ready to be viewed.

Sincerely,
Michela Gambino
Editor
mSystems